# Sardine: A modular framework for developing data acquisition and near real-time analysis applications

A. Lucas Martins, Alexandre Laborde, Michael B. Orger*

Champalimaud Research, Champalimaud Foundation, Lisboa, Portugal

* michael.orger@neuro.fchampalimaud.org

## Abstract

We present Sardine, a software framework built with .NET to control experimental setups through the reliable execution of dynamic networks of independent modules, where each module can interface with a hardware device (e.g., camera, motor) or represent an operation over data (e.g., image filter, data stream). The Sardine framework delivers robust fault-tolerant hardware control and eliminates downstream delays by implementing dedicated data processing queues for each module. Any .NET class can be seamlessly adapted into a Sardine module, enabling effortless integration with existing codebases. Sardine's modular architecture ensures flexibility to accommodate changes in experimental paradigms, simplifying the adaptation of essential features, such as swapping hardware components or redefining the logic of their interactions. The core of Sardine is a novel aggregation system that connects modules in two layers, designed to streamline complex workflows such as those in microscopy applications. The first layer (link layer) enables concurrent operations between modules, such as synchronizing a camera module with a stage controller or a laser module, while maintaining a dependency tree to ensure devices operate as intended. For example, in an imaging experiment, Sardine oversees individual module malfunctions, such as a camera failure or stage misalignment, gracefully handling errors and dynamically restoring functionality to impacted network segments to minimize disruption. The second layer (data layer) facilitates the transmission of information across modules by associating them with operations that produce, consume, or transform data. In an imaging context, this could involve processing raw image data from a camera module, passing it through a real-time analysis module for feature detection, and forwarding the results to a visualization module for immediate feedback. This two-layer architecture ensures seamless data flow and robust error handling, making Sardine ideal for dynamic and complex experimental setups. Sardine also integrates logging, a metadata collection system, and tools to create graphical applications.

**Data availability statement:** The source code of Sardine is available at https://github.com/orger-lab/sardine. A gallery of components written for use with Sardine and examples are available at https://github.com/orger-lab/sardine-components. Pre-built packages of Sardine can also be downloaded from NuGet at https://www.nuget.org/packages/Sardine.Core and https://www.nuget.org/packages/Sardine.Core.Views.WPF.

**Funding:** This work was financed by funding to Michael Orger's lab from the European Research Council (Consolidator Grant, Neurofish-DLV-773012), the Champalimaud Foundation, the Volkswagen Stiftung Life? Initiative (A126151) and through national funds from the Portuguese FCT - Foundation for Science and Technology, I.P., under the projects PTDC/NEU-SCC/5221/2014 and 2023.14873.PEX. A. Lucas Martins received funding from the Portuguese Fundação para a Ciência e Tecnologia (FCT) through the PhD fellowships SFRH/BD/129843/2017 and COVID/BD/152726/2022.

## Introduction

Many modern neuroscience research questions rely on multimodal experimental setups, that collect data from multiple sources and control diverse hardware setups, such as electrophysiology rigs, optogenetic stimulation systems, or behavioral tracking devices. Custom-built solutions are often hard to expand, replicate, or adapt to new paradigms, especially those incorporating real-time closed-loop feedback. These challenges highlight the need for flexible, modular frameworks that can seamlessly integrate and control diverse hardware and software components while enabling reproducibility and adaptability across experiments.

Developing near real-time acquisition and control systems is technically demanding: software must run multiple sources and processes in parallel, handle asynchronous data streams, and meet strict timing constraints. Furthermore, software that operates an experimental setup is likely to change over time, motivated by short-term experimental requirements. Particularly when the developer is also the end-user, changes may be made quickly to provide a necessary feature without regard to the long-term stability of the code [1], leading to an accumulation of technical debt [2] that makes systems increasingly hard to debug, adapt and reuse.

Software orchestration solutions can be employed to manage interactions in dynamically coupled software systems, handle communication between components, and safeguard the system's status. Most existing solutions are focused on managing modularity for distributed or large-scale systems, such as cloud computing or containerized services [3], and they are not tailored to the specific needs of experimentalists who need near real-time control on local desktop machines.

Numerous practical guides have been written that list recommendations guiding the research developer in the direction of generating quality software [1,4–8]. Many recommendations reference the application of Agile principles [4,9,10], which are keystones of efficient software development in industry. These require the user to be able to create reusable and pragmatic code structures. Other common recommendations are producing quality documentation, designing graphic user interfaces, using version control tools, and building quick-start guides and tutorials. Several working groups have collected standard practices of successful software, defined the concept of software sustainability, and designed guidelines for creating sustainable software for research [11–13]. For example, Connolly and others have defined three main characteristics a software solution should achieve [11]: readability, written to promote understanding by others; resilience, a system that fails rarely and that fails gracefully; and reusability, a system whose parts are modular and can be easily applied in different scopes.

Most researchers are not software developers, and are likely to prioritize progress towards their experimental results rather than adherence to coding best practices. They would benefit from tools that make well-structured, reusable software the default outcome, rather than an additional burden.

Motivated by this need, we have developed Sardine (Fig 1), an open-source framework written in `C#` and built with `.NET 8` [14], for rapidly creating and execute

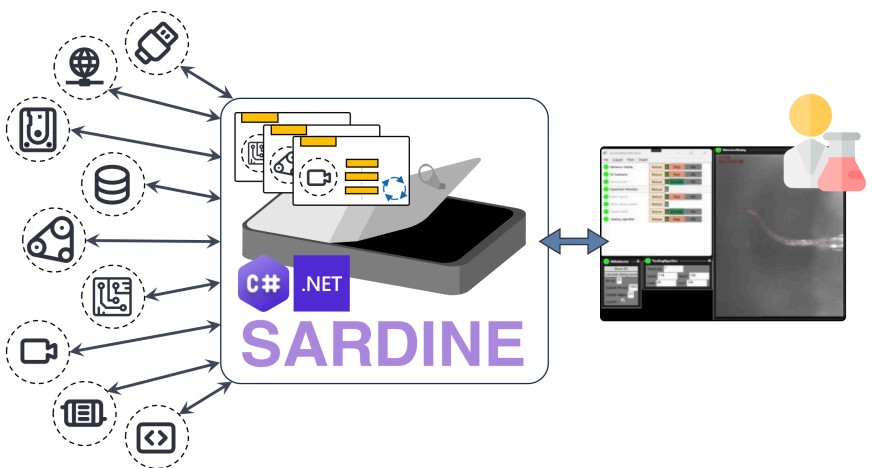

**Fig 1**. **An overview of the Sardine framework.** Sardine is an open-source framework written in `C#` and built with `.NET 8`. It is designed for developing and executing complex scientific workflows. This framework allows users to quickly create and run data acquisition and processing desktop applications using a modular approach. In Sardine, complex systems are divided into components that are developed separately and encapsulated into modular units called vessels. The framework orchestrates the coordinated operation of these vessels.

modular desktop applications for data acquisition and processing. In Sardine, complex systems are broken down into components that are developed separately, and the framework orchestrates their conjoined operation. Sardine identifies module failures, provides fault-tolerance mechanisms to keep an updated system status, and supports highly concurrent data flows between modules while keeping a low overhead. Sardine also supports the integration of graphical user interfaces, including a launcher, a component manager, and prototypes for user controls.

In Sardine, every hardware component or data operation is encapsulated within its self-contained environment. Building an application simply involves defining these components and specifying their relationships, while the framework's underlying architecture automatically manages the rest, ensuring seamless integration and operation.

Components are connected in a novel two-layer network represented by two graphs: one defining hierarchical dependencies and the other defining data flows. By separating connections into critical dependencies and non-critical messaging, a system built with Sardine can operate even in the presence of perturbations, as malfunctioning components and their hierarchical dependencies can be safely terminated by the framework.

Sardine is designed to naturally produce sustainable codebases that are highly reusable, easily maintainable, and capable of quickly producing functional prototypes. Using Sardine, every component and connection between components is built and tested independently, increasing code resiliency and enforcing separation of concerns. Sardine natively handles concurrent execution, manages data operations for each component, ensures the correct execution of operations, and manages logging.

We envision this framework as a tool for those who develop for the research environment, from the in-house software engineer building and maintaining several setups to experimentalists building their own systems that they are interested in sharing with others.

We have used our framework to develop solutions for the control of data acquisition and stimulation systems in behavioral neuroscience, demonstrating its technical capabilities. For example, we have built desktop applications capable of performing online posture tracking of zebrafish larvae at speeds of 700 Hz, a critical feature for studying fast-moving organisms and capturing subtle behavioral changes in real time. We have also developed visual stimulation schemas with closed-loop feedback latencies faster than the refresh rate of the display, enabling precise real-time experimental

paradigms, such as those requiring immediate adjustments based on live data, essential for studies involving neural feedback loops or dynamic stimuli. Additionally, we have created control systems for custom-built microscope setups that acquire and process data at bandwidths of up to 1600 MB.s⁻¹, showcasing the framework's ability to manage high-speed, high-bandwidth data flows necessary for advanced imaging techniques and large-scale data analysis. These examples highlight the framework's capacity to address critical challenges in neuroscience research, empowering researchers to implement complex experimental designs with precision, scalability, and reproducibility.

## Methods

### The functional organization of Sardine

The functionality of an acquisition system can be defined as a collection of distinct components and their expected interactions. Each component may represent, for instance, a hardware device (e.g., camera, motor, serial port) or an operation over data (e.g., image filter, data stream, waveform).

The software that controls this system will contain instructions to execute the functionalities of each of its components. In an object-oriented programming approach, system components can be seen as objects represented in code by classes, originating in a publicly available API, such as a controller made by a hardware manufacturer or code written by the system developer.

To characterize the communication between a pair of components $(c_1, c_2)$, we distinguish between two types of dependencies between those components: if there is one component $c_2$ which can only function correctly if $c_1$ is functional, then a strong hierarchical dependency exists between that pair. For example, a camera $c_2$ may only be able to run if its corresponding controller board $c_1$ is active. On the other hand, if both components can function independently from each other, they have a weak dependency between them. This is what happens when transmitting acquired data between components. For example, a video recorder that saves images from an array of cameras might keep working even if one camera malfunctions.

In Sardine, access to each system component is encapsulated in a **vessel**. Vessels form the modular units of the framework and interconnect to create two distinct networks: the **link graph** and the **data graph**. The former represents the hierarchical dependencies between components, while the latter defines the messaging system. The set of all vessels and their connections is called a **fleet**, representing the system to be executed. Each vessel simultaneously functions as a node in both graphs and can participate in each in different ways.

The vessel's primary function is to maintain a state machine that keeps track of the status of the underlying component and guards against unexpected side effects, thereby protecting the other vessels, even if they are interconnected. To achieve this, it is loaded with instructions on how to instantiate and start the underlying component and how to terminate its execution correctly and dispose of it at termination. A pair of control variables keeps the current state of each vessel, with one variable (`IsOnline`) indicating if the underlying component is running and another representing if the vessel is active in data communication (`IsActive`), which the user controls. Furthermore, to safeguard the vessel against potential failures resulting from external utilization of the component, an `Execute Call` method has been introduced, which serves to encapsulate any external application of component code. The control flow of a vessel is schematized in Fig 2.

Hierarchical dependencies between vessels are defined during the vessel creation. They are used to form the link graph, a directed acyclic graph where nodes represent vessels and edges represent dependency relations. When the user runs an application built with Sardine, the link graph is automatically generated and topologically ordered, and components are instantiated according to their rank: later components only start if earlier components are already running. For example, in Fig 3, the components of vessels **A** and **F** will start first, followed by **B** and **G**, and Sardine will only turn on later components if earlier components start successfully.

If Sardine can't access the underlying component of a vessel or an exception is thrown during a call to one of its methods, the vessel will automatically turn off. Malfunctions in one vessel do not cause the entire network to fail; instead, they

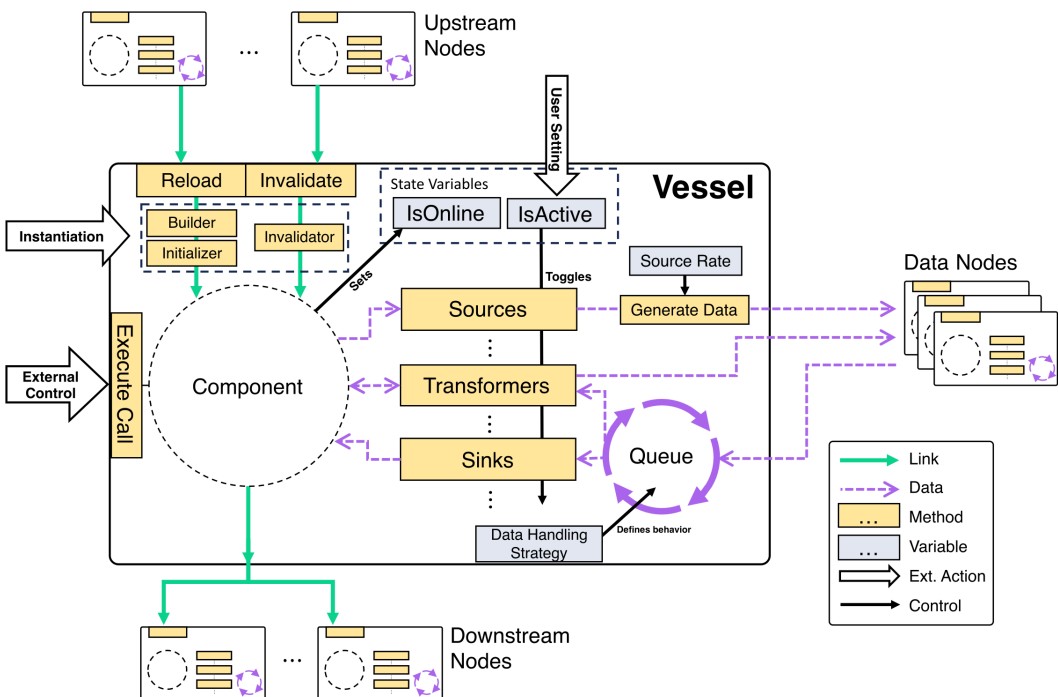

**Fig 2. Diagram of a vessel.** In Sardine, system components are adapted into a vessel, providing components with extra features while safeguarding their state. The Freighter class is responsible for instantiating Vessels. The Vessel class has the `Builder`, `Initializer`, and `Invalidator` methods. The builder is responsible for creating a component object, and the initializer is responsible for setting up the component after its creation. The invalidator properly marks the component as invalid in cases of termination or failure. A vessel may depend hierarchically on other vessels, meaning that the creation of its component requires access to other components, which must be properly initialized beforehand. These correspond to upstream nodes in the link graph. Conversely, a vessel is a dependency on downstream nodes in the link graph. The component's state (online or offline) is observable through the `IsOnline` state variable. A wrapping call called `Execute Call` is provided to access the component code properly. Any data operations may be associated with a vessel - sources, transformers, or sinks, depending on whether they produce, modify, or consume data, respectively. The control variable `IsActive` toggles the data operations on and off. Sources are triggered by the `Generate Data` method, which can be polled periodically (at the rate defined by the `Source Rate` variable) or at will by the user. Incoming data is either immediately processed, or placed in a processing queue. The `Data Handling Strategy` variable defines this behavior. The link graph is represented by pink arrows, and blue dashed arrows represent the data graph. Vessel methods are represented by yellow boxes, and light green boxes represent vessel control variables. External actions are represented by large block arrows, and internal control flow by black arrows.

generate a signal that propagates downstream through the link graph, shutting off any vessels that are dependent on the malfunctioning one. In Fig 3, we exemplify the failure of a component in vessel **B**. While that causes a breakage, it is locally contained, and the remaining network keeps functioning. The malfunctioning vessel can be dynamically regenerated at any moment, repairing the graph. The vessels' state is managed through the reload and invalidate methods. The reload method is responsible for constructing and initializing the vessel component and switching the vessel state to online. In contrast, the invalidate method properly marks the component as invalid, terminates the component, and changes the vessel state to offline.

When starting an application, Sardine calls the reload method of all vessels, following the topological order of the link graph. Users can also invoke the reload method at will to reset the component. The invalidate method is called automatically whenever a failure occurs within the component or the vessel, which also triggers the invalidation of downstream nodes.

This robustness is a hallmark of Sardine: it prevents the system from running in an unstable status, where the produced behavior might be different from what is expected, unbeknownst to the experimenter. For example, it stops parts

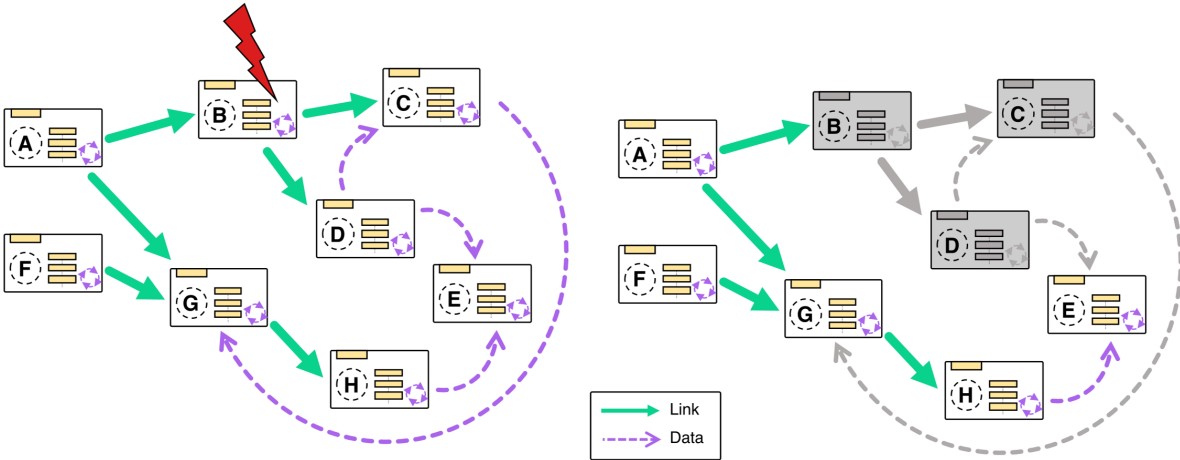

**Fig 3**. **A representation of a fleet.** Vessels labeled **A** through **H** are interconnected through hierarchical dependencies, which are illustrated by the pink arrows in the link graph and by data operations, shown with dashed blue arrows in the data graph. A malfunctioning component does not lead to the failure of the entire application. *Left:* the fleet is in a normal state when vessel **B** malfunctions. *Right:* the fleet's status after the malfunction. In this scenario, vessel **B** and its downstream nodes **C** and **D** are invalidated, but the remaining network continues to operate normally. E can still operate over data that arrives from **H**.

from receiving incorrect inputs due to the malfunctioning of an upstream component. Simultaneously, it ensures no experimental data is wasted. In a system where several sources are acquiring data concurrently, the failure of one of the source components will not impede the other parallel workflows from continuing to function normally. It also helps ensure effective debugging of the system, as one can quickly identify breakage points of the system.

## Vessels communicate asynchronously through messages

Sardine defines a data model that enables any number of data operations to be associated with a vessel, generating new data or operating over data generated by another vessel. Data produced through a data operation is packed into a message with metadata, including timing and source information, and automatically made available through the network. Sardine operations are of three types, depending on their data flow: sources that generate new data (e.g., a camera producing an image frame), transformers that operate over data and create a modified version of it (e.g., a tracker that extracts the coordinates of a sample from an image), and sinks that consume data (e.g., a recorder that saves sets of coordinates to a text file). The following metadata is associated with each payload: the types of data produced, the vessel that sent the message, the original source of the message, the polling rate of the message source, and a message identifier number.

At runtime, the framework generates the data graph. For each data operation associated with a vessel, all other data operations that could provide input to it are selected, and a directional asynchronous communication channel is created as an event callback. These connections form a weakly referenced multidigraph: if one vessel is unavailable to receive or send messages, the remaining network keeps operating normally (Fig 3). Vessels have a `Generate Data` method that, when invoked, initiates the creation of new messages from available sources. Vessels that include data sources can be regularly polled by Sardine. Alternatively, data generation can be triggered by the user at chosen moments. Data is, by default, multicast to all vessels with a data operation that can process it. Otherwise, source filtering can be applied to each data operation that receives data so that only messages of interest are processed.

While most of the load of a modern application is automatically managed by the operating system and the processors they run on, desktop computers do not run in real-time, meaning that there isn't a fixed deadline for executing a given

operation — they are not deterministic. In practice, while estimating the duration of an operation is possible, one cannot assure messages will arrive at constant times to vessels downstream, nor can one ensure they will be processed in the same time window [15]. Sardine vessels work equivalently to single-server networks, where implementing a queue protects against the variability in message arrival and processing time.

Each vessel has access to an inbox queue, where incoming messages are placed before processing. Data is read in a first-in/first-out format, assuring messages arriving at a vessel are processed sequentially. All processing queues and process operations work in parallel and are non-blocking, avoiding the need for constant polling for a low computational overhead. The main aim of the inbox queue is to protect upstream vessels against processing delays, which are expected to happen due to the uncertainty associated with task scheduling. The inbox queue also dampens any temporary delays caused by vessel creation or their dynamic reloading. The size of this queue is defined at vessel creation, and the user can observe the queue status at any time.

Furthermore, since messages are placed in the inbox queue before processing, if multiple vessels try to communicate with the same downstream node, their messages will be added to the queue and processed sequentially, avoiding collisions in accessing reserved resources. Furthermore, since all messages carry a unique ID, messages can be uniquely resolved and ordered by the user as appropriate if such behavior is required.

It is also possible to opt out of using the queuing behavior and instead have the incoming message processed directly. This allows for faster execution of instructions at the expense of possible upstream delays and having to manually handle possible collisions.

### Services provide extra flexibility to Sardine

Sometimes, a process would benefit from arbitrarily accessing any system component occasionally and for short periods, for which creating the necessary data operations would be cumbersome. Consider a process that collects execution information and metadata from components to generate a report. This report might be made once during the whole experiment, and the computing effort of this task is low compared to regular system operation.

To support processes that randomly access component information and can bypass the data messaging system, Sardine implements support for **services** — extensions that can be associated with a fleet to provide it with new functionalities. Services are lightweight wrappers used for simple tasks and are accessible to all other system components.

Contrary to vessels, services do not participate in either the link or the data graph. Instead, they are kept in a services pool that is associated with the fleet. When a service is called for the first time, an instance of the underlying process is created and maintained in the services pool associated with the fleet. That instance is reused on subsequent calls, maintaining its state throughout the operation of the application.

The core implementation of Sardine includes several support classes meant to be used as services: a file manager, a metadata collection system, and a remote controller. The file manager provides input files to the application, such as setting files or experimental protocols, through user-built file handlers, and the UI management system uses it to load application layouts. The metadata collection system provides a mechanism for collecting parameters and settings from vessels according to user specifications. Finally, the remote controller extends vessels to provide a mechanism to associate text-based commands to execute arbitrary code and exists to provide support for interprocess communication tools.

### Sardine produces user-friendly desktop applications

One crucial consideration for developers is including functional control interfaces in their software. Developing good GUIs (graphical user interfaces) can be complicated, especially when creating reactive elements that respond promptly to user commands. To tackle this, Sardine includes a collection of UI (user interface) design assets built with WPF (Windows Presentation Foundation) that allows the launch of any fleet description as a full desktop application.

A UI model provides scaffolding to create event-driven user controls for vessels. Through the UI model, a custom view interfaces with the vessels through an intermediate view model generated automatically at runtime by Sardine. This intermediate view model inserts hooks into the public properties of the components for seamless control with callbacks and solves most of the complexity of assuring a correct separation between user controls and the runtime logic of the components. The underlying design pattern also forces the design constraint that a view must be built for each component separately, ensuring modularity at the user control level.

Views are automatically assigned to corresponding vessels at runtime, leveraging the `.NET` reflection tools, which allow dynamic access to existing assemblies and types. Two notable aspects of Sardine views are that a vessel may have more than one associated view, which the user can select according to their needs, and each view runs independently of the others on a separate thread. If a vessel or its view stalls, all others keep operating normally.

Several helpful user controls are available by default when running a Sardine application. The default view of the application provides information regarding the status of the system and allows control over the state of the vessels (Fig 4). This view also shows information about the occupation of input queues to each vessel. Finally, the application's main window can be extended through UI helpers, which are added to the fleet manager as option menus and provide extra details on the system's status. Some helpers are included by default, such as a graph visualizer that shows the network of all vessels, a logger window that allows quick access to the system's debug information, and a simple file manager. Custom view layouts can be built from the same fleet, which can be saved and imported back. For example, the same experimental setup may be used differently depending on the user and the experimental assay, requiring interaction with different components. A custom layout allows for a cleaner, less obtrusive visual design, lowering the chances of accidental mistakes due to configuration errors.

## Operation

To use Sardine in your `.NET` application, download and install the packages via the `NuGet` package manager or the `NuGet` command line interface.

```
1  nuget install Sardine.Core
2  nuget install Sardine.Core.Views.WPF
```

### Creating a Fleet and encapsulating components in Vessels using the Freighter

A class must be written to represent each component. Any `.NET` class can be wrapped into a vessel, and no particular considerations or rules must be followed for compatibility with Sardine. Therefore, existing codebases or third-party APIs written in `.NET` can be adapted readily.

This section provides an overview of how Sardine is used to create an application from scratch. We will describe a reduced example system and build a control software using Sardine to show how the framework operates: a camera records a scene, and images from the camera are saved to disk. Consider that the manufacturer provided the camera API, where the camera is controlled via a `Camera` object, which is, in turn, produced by a `CameraFactory`. Also, consider that the code that saves images to disk exists within the class `DataSaver`.

```
1  // Example class signatures
2
3  // This class contains an empty constructor, and a GetCameras method that returns a list
       of Camera objects
4  public class CameraFactory
5  {
6      public CameraFactory();
```

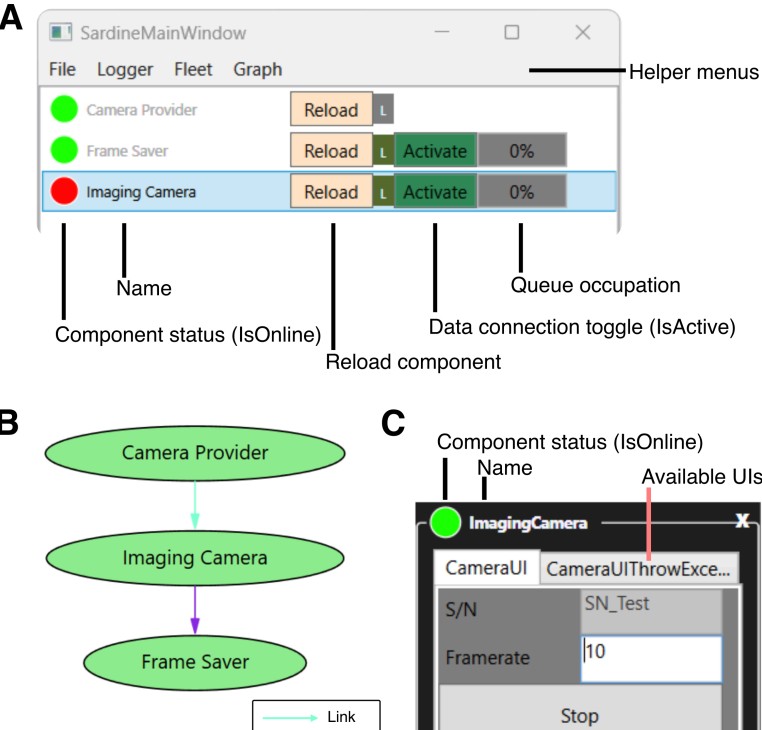

**Fig 4**. **An example Sardine application running on Windows. A** The application's main window displays a list of all vessels, providing information on their status and access to reload methods, as well as control over data connections. **B** The fleet graph diagram, as generated by Sardine. The link graph edges are represented by blue arrows, and the data graph edges are represented by purple arrows. **C** An example control displays various user interfaces for one vessel through a tabbed layout.

```
 7      List<Camera> GetCameras();
 8  }
 9
10  // The Camera class can only be created by CameraFactory. It contains a method
        GetNextFrame that returns a frame if one is available, start/stop methods, and a
        controllable frame rate
11  public class Camera
12  {
13      public string SerialNumber { get; }
14      public int FrameRate { get; set; }
15
16      public void Start();
17      public void Stop();
18      public CameraFrame? GetNextFrame();
19  }
20
21  // CameraFrame object wraps a byte array containing the camera data
22  public class CameraFrame
```

```
23 {
24     byte[] Data { get; }
25 }
26
27 // Data saver is able to save a byte array to disk, through its method SaveData
28 public class DataSaver
29 {
30     public DataSaver(string path);
31     public void SaveData(string filename, byte[] data);
32 }
```

A Sardine application requires the creation of a `Fleet` object that represents the system. A new class of type `Fleet` is to be created, and that object must contain a public property of type `Vessel<TComponent>` for each system component.

In the constructor of the `Fleet` object, vessels will be instantiated using the Freighter class, which packages any given `.NET` class. In this example, we build a fleet that represents the system mentioned above. We define builder methods for `CameraProvider` and `FrameSaver` using the constructors of their underlying class. For the `ImagingCamera` vessel, a dependency on `CameraProvider` is declared since the underlying `Camera` object will be extracted from it by the builder method. We show an example of where a camera with a specific serial number is selected from all available cameras. In this case, we include an initializer method, which will be run immediately after building, calling the `Start` method on the underlying `Camera` object. We also include an invalidator method, which will be called at program termination, by the user turning off the `ImagingCamera` vessel, or in case of an error.

```
1 using Sardine.Core;
2
3
4 namespace ExampleSystem;
5 public class MySystem : Fleet
6 {
7     public Vessel<CameraFactory> CameraProvider { get; }
8     public Vessel<Camera> ImagingCamera { get; }
9     public Vessel<DataSaver> FrameSaver { get; }
10
11     public MySystem()
12     {
13
14         CameraProvider = Freighter.Freight(
15         builder: () => new CameraFactory()
16             );
17
18         ImagingCamera = Freighter.Freight(
19         CameraProvider,
20         builder: (provider) => provider.GetCameras().
21             Where(camera.SerialNumber == "CAMERASN").FirstOrDefault(),
22         initializer: (provider, camera) => camera.Start(),
23         invalidator: (provider, camera) => camera.Stop()
24                         );
```

```
25
26          FrameSaver = Freighter.Freight(() => new DataSaver("images"));
27      }
28 }
```

### Adding a data operation

One must design the corresponding data operation methods to route data collected by the camera to the saving schema, following one of the supported signatures.

```
1 public delegate TOut? Source<out TOut>(THandle handle, out bool hasMore);
2 public delegate TOut? Transformer<in TIn, out TOut>(THandle handle, TIn data,
     MessageMetadata metadata);
3 public delegate void Sink<in TIn>(THandle handle, TIn data, MessageMetadata metadata);
```

In the current example, we write a `Source` method that will be attached to `ImagingCamera` and a `Sink` method that will attach to `FrameSaver`. For the former, a frame is collected from the camera, and a check is done to see if it was collected successfully, in which case we set `hasMore` to `true` so that the sourcing will happen again. For the latter, binary data is extracted from an incoming camera frame and saved using `DataSaver`. Some of the auto-generated metadata is used to establish a name for the saved file. Note that access to the source code of the underlying components isn't required to write or apply the data operations.

Then, we hook the methods to the existing `Vessel` objects and activate Sardine's automatic data generation feature by setting the polling rate of `ImagingCamera`.

```
1 public static CameraFrame? CameraFrameSource(Camera cameraHandle, out bool hasMore)
2 {
3     CameraFrame? frame = cameraHandle.GetNextFrame();
4
5     hasMore = (frame != null);
6
7     return frame;
8 }
9
10 public static void FrameSink(DataSaver saverHandle, CameraFrame dataIn,
11   MessageMetadata metadata)
12 {
13     string filename = $"{metadata.SenderName}_{metadata.SourceID}";
14     saverHandle.SaveData(filename, dataIn.Data);
15 }
16
17 ---
18
19 public MySystem()
20 {
21 ...
22 ImagingCamera.AddSource(CameraFrameSource);
23 FrameSaver.AddSink(FrameSink);
```

```
24 ImagingCamera.SourceRate = 100;
25
26 ...
27 }
```

**Building a desktop application**

Windows desktop applications can be created from fleets by using Sardine's WPF hooks. From a desktop application project targeting `net8.0-windows`, with WPF enabled, all that is needed is to modify the startup object to use Sardine. At execution, Sardine will create an instance of the fleet, generate the corresponding graphs, and build all vessel components. By default, the logging service is already running, and the Fleet Manager UI opens, letting the user control the status of the components. The system is fully operational, and the user can activate components anytime.

```
1 --- XAML
2 <sardine:SardineApplication
3 xmlns="http://schemas.microsoft.com/winfx/2006/xaml/presentation"
4 xmlns:x="http://schemas.microsoft.com/winfx/2006/xaml"
5 xmlns:fleet="clr-namespace:ExampleSystem;assembly=ExampleSystem"
6  xmlns:sardine="clr-namespace:Sardine.Core.Views.WPF;assembly=Sardine.Core.Views.WPF"
7 x:TypeArguments="fleet:MySystem"
8 x:Class="ExampleApplication.App"
9 />
```

```
1 --- Code behind
2 using Sardine.Core.Views.WPF;
3 using ExampleSystem;
4
5 namespace ExampleApplication;
6
7 public partial class App : SardineApplication<MySystem> { }
```

**Adding a GUI**

We show how the Sardine UI Model can quickly connect a custom GUI to a component. By inheriting from the `VesselUserControl<T>` class, one can interface a WPF user control with a vessel of the same underlying type. The example shown is a user interface for the `Camera` class that allows the user to set the `FrameRate` variable and access the `Start` and `Stop` methods. Since Sardine automatically manages user controls via reflection, if the assembly containing the user control is referenced in the application project, there is no need for further integration. For the sake of simplicity, only the markup meaningful to the control elements that regard the framework is shown. Note how a component property can be accessed directly using WPF data bindings. At runtime, Sardine automatically creates the view model and its bindings, interfacing the text box value with the component property and hooking the `PropertyChanged` callback to observe changes in the underlying component, making the user control reactive. Users can further customize their GUIs by using any natively available WPF resource or UI control, as well as any of the many third-party libraries available online.

```
1 --- XAML
2 <sardine:VesselUserControl
```

```
3      x:Class="ExampleApplication.CameraUI"
4      x:TypeArguments="fleet:Camera"
5      xmlns="http://schemas.microsoft.com/winfx/2006/xaml/presentation"
6      xmlns:x="http://schemas.microsoft.com/winfx/2006/xaml"
7      xmlns:local="clr-namespace:ExampleApplication"
8      xmlns:sardine="clr-namespace:Sardine.Core.Views.WPF;assembly=Sardine.Core.Views.WPF"
9      xmlns:fleet="clr-namespace:ExampleSystem;assembly=ExampleSystem"
10     ...
11 >
12 ...
13 <TextBox Text="{Binding Path=FrameRate, UpdateSourceTrigger=PropertyChanged}"/>
14 <Button Content="Start" Click="ButtonStart_Click"/>
15 <Button Content="Stop"  Click="ButtonStop_Click"/>
16 ...
```

```
1 --- Code behind
2 ...
3 public partial class CameraUI : VesselUserControl<Camera>
4 {
5      public CameraUI ()
6      {
7          InitializeComponent();
8      }
9
10     private void ButtonStart_Click(object sender, RoutedEventArgs e)
11     {
12         Handle?.Start();
13     }
14
15     private void ButtonStop_Click(object sender, RoutedEventArgs e)
16     {
17         Handle?.Stop();
18     }
19 }
20 ...
```

Any other event callback can also be associated with a vessel property by using `VesselPropertyToEventDependencyObject`:

```
1  <sardine:VesselUserControl.EventUpdatesCollection>
2   <sardine:VesselPropertyToEventDependencyObject PropertyName="Name" EventName="
     OnThisEvent"/>
3 </sardine:VesselUserControl.EventUpdatesCollection>
```

### Adding a service

Services are unique instances created when first invoked and associated with the fleet. Any `.NET` class with an empty constructor can become a service via a call to `Get<TService>()`. Service access can only be done sequentially. As such, they are not prepared to carry out demanding computational tasks, and trying to do so might cause a component to be blocked while waiting for access to the service. In addition, debugging their interaction may prove more difficult. The upcoming example will illustrate these principles by showcasing a service designed to interact with an external system (such as a remote server or an external logging service), demonstrating how services can signal state changes in a controlled manner by signaling whenever the `ImagingCamera` vessel is turned on or off.

```
1  // Mock-up of the external messaging class
2  public class ExternalMessaging
3  {
4    public ExternalMessaging();
5    public void SendMessage(string message);
6  }
7
8  ...
9  // The ImagingCamera initializer and invalidator is
10 // changed to add a service call
11 ImagingCamera = Freighter.Freight<Camera>(
12   CameraProvider,
13   builder: (provider) => provider.GetCameras().
14     Where(camera.SerialNumber == "CAMERASN").
15     FirstOrDefault(),
16   initializer: (provider, camera) =>
17     {
18       camera.Start();
19       Get<ExternalMessaging>().
20         SendMessage("Camera on!");
21     },
22   invalidator: (provider, camera) =>
23     {
24        camera.Stop();
25       Get<ExternalMessaging>().
26         SendMessage("Camera off!");
27     },
28 );
29 ...
```

### Layout and control of a Sardine application

At the start, a loading screen appears while the application components are being built. Once this process is complete, the main window of the Sardine application opens, displaying a list of all vessels in the active fleet (Fig 4).

To the left of each vessel's name, a colored circle indicates its status using a semaphore light scheme: red for offline, yellow for loading, and green for online. On the right side of the vessel name, there is a button that toggles the vessel

between inactive and active status (`IsActive`). An active vessel participates in the data graph, while an inactive vessel neither sends nor receives messages.

Next to the vessel name, a percentage indicator shows the status of that vessel's inbox queue. Double-clicking a vessel name opens all available user controls in a tabbed view. If a vessel's name is grayed out, it means that there are no user controls available for that vessel.

Right-clicking on a vessel's user control reveals additional options, such as locking the position of that UI element on the screen or changing the visibility of the tabbed controls. The top menus provide access to other functionalities, including the file manager, the logger, and a view of the fleet graphs.

Under the File menu, the "Save Layout" option records the screen positions of all user layouts along with the values of set properties, creating a file that can later be opened and recalled.

### Using the settings manager

At the beginning of execution, Sardine will read and use the settings file `SARDINE.xml` to define its behavior, which is stored in the same folder as the compiled executables. Settings can be accessed via the `FetchSetting` and `FetchSettings` methods (default settings are listed in Table 1. The example shows how a variable in the fleet `MyFleet` is defined according to a field of the settings file:

```
1 --- SARDINE.xml
2 <SARDINE>
3   <CameraSettings CameraSN="SN_Test"/>
4 </SARDINE>
```

```
1 --- MyFleet
2 public MyFleet(){
3 // Retrieve the camera serial number from settings
4 var setting = Current.SettingsProvider.FetchSetting("CameraSettings", "CameraSN");
5 string cameraSN = null;
6 // Check if the setting is not null before accessing its value
7 if (setting != null)
8     cameraSN = setting.Value;
9 ...
10 }
```

**Table 1**. **Sardine settings available by default.** The settings name column is the XML path where that setting is stored.

| Setting Name | Description |
|---|---|
| UserDataPath | Directory where user data and logs are saved |
| Name | Application name |
| HelpersMenu → ShowHelpers | If the menu with helpers should be displayed (True/False) |
| Application → GenerateSardineWindow | Whether the main Sardine window should open (True/False) |
| Application → Loader → BorderBrush | Color of the loading screen border brush |
| Application → Loader → SplashImagePath | Image to show during the loading sequence |
| Application → Layout → Default | Default UI layout to open after loading |
| Application → HelpersMenu → Helper → Name | Helper menus to be shown |
| Application → Layout → Window → Type | Windows to open after loading (other than the main window) |

### Using the logger

The logger collects messages generated by Sardine and all vessels. Sardine logs the status changes of vessels and any errors that occur during operation. By default, all standard output generated by components is rerouted to Sardine's logger and saved, and this behavior is controlled by the vessel's `CaptureLogs` property. The default logger uses log messages with eight different levels of importance. Custom log messages can be sent using the available logging extensions.

```
1  \\ Logging a message originating from a vessel
2  MyVessel.Log("This is a custom log message.", LogLevel.Information);
```

### Extending the Sardine application through UI helpers

Using UI helpers, new menus and options can be added to the Sardine application. To create a new UI helper, one must implement the `IUIHelperProvider` interface. At runtime, Sardine finds all existing implementations of `IUIHelper-Provider` via reflection and automatically builds the user interface menus.

### Use cases

#### An online tracking system for head-fixed larval zebrafish

We describe a system designed for tracking tail features in head-fixed larval zebrafish during in-vivo preparations within virtual reality assays. This system must then collect images from a high-speed camera, process each frame to extract information related to the larva movement between frames, and update a rendered scene so it reflects the change in the virtual reality environment in response to the larva movement. To ensure the success of this online tracker, we identified three main requirements: the median processing time for each camera frame must be bellow approximately 1.43 ms, as the system requires a collection rate of 700 Hz; the processing queue must be able to hold sufficient input data to minimize the likelihood of it filling completely; and imaging data must be processed continuously to extract behavioral metrics, which can then be used in feedback control for stimulation. This simple tracker is efficient, computing tail posture in under 150 microseconds per frame, and making it robust to other stresses on system performance

This software was adapted and used in different setups, with distinct camera hardware and computer specifications, and has been successfully used to produce data for several publications [16–18].

A modified version of this software, *VirtualFishTracker*, is provided as an example. To build this example, **Windows 10 version 1607 or above** and  **Visual Studio version 17.8 (2022) or above** are required. Full instructions are available in the https://github.com/orger-lab/sardine-components/ repository. The hardware components have been replaced with mock versions that use data from a movie recorded with one of the experimental setups.

The system is organized into several components: a camera, a camera controller, a background subtraction algorithm for the camera feed, a tail tracking algorithm that operates on the background-subtracted images, a display for visualizing both the camera feed and the tracking results, a model that generates feedback signals based on the tail tracking results for visual stimulation, a set of recording components capable of saving both a binary stream of camera images and the calculated metrics, and an experiment control utility (Fig 5).

Each component was implemented independently and integrated using Sardine. We also developed user controls to facilitate end-user system operation. The background subtraction component utilizes OpenCV runtime libraries to perform image manipulation, while the display makes use of the SkiaSharp library to render visual elements on the screen. The experiment control utility allows users to specify a folder for recording experiment data. Also, it creates a file containing metadata about the executing components and other relevant information regarding the experiment. This ensures that essential acquisition details are stored alongside the acquired data in a human-readable format, allowing for easier verification.

**Fig 5**. **The fleet graph diagram of the *VirtualFishTracker* application**. Sardine automatically generated this graph. The link graph edges are represented by blue arrows, and the data graph edges are represented by purple arrows.

Opening the application launches its GUI (Fig 6). Activating the `Mock Camera` instantly begins data production. Double-clicking the vessel name `Display` opens the display, which renders images once it is turned on. To initiate tracking, both the `BG Subtractor` and the `Tracking Algorithm` must be activated. Activating the `Output Writer` starts recording tracking results to a text file, while the `Binary Writer` captures images in a binary stream. It is important to note that once the writers are activated, they will automatically record to the folder containing the application's executable file.

The `Experiment Metadata` vessel can be used to specify a recording path. It leverages Sardine's metadata tools to gather acquisition parameters, including tracking and background subtraction options, and configure the output path for

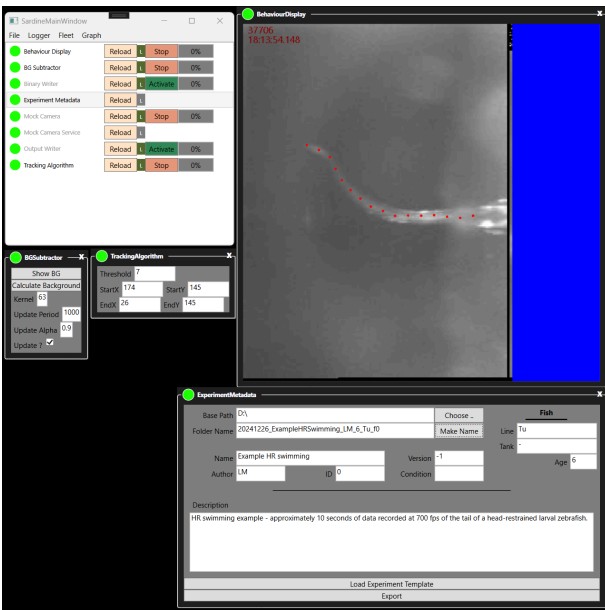

**Fig 6**. **A screenshot of the *VirtualFishTracker* application**. The main window is visible, together with a set of UI components that are associated with vessels. The display shows data that is being transmitted as messages, originating from the video source and the tail tracker.

the recorders. Our `Experiment Metadata` vessel also supports the inclusion of metadata related to experimental conditions. Pressing the Export button creates an output folder structure to organize the acquired data and establish the paths for the recorder vessels.

The user protocol for using this software might look like this:

1. Activate the `Display`, `Mock Camera`, `Tracking Algorithm`, and `BG Subtractor`.
2. Set the parameters for `BG Subtractor` and `Tracking Algorithm` to achieve satisfactory tail tracking.
3. Specify the experiment metadata parameters, such as experiment name and animal ID.
4. Export the metadata.
5. Reload the `Mock Camera`.
6. Activate the `Binary Writer` and `Output Writer`.
7. Activate the `Mock Camera`.

This sequence of actions allows the user to set the correct parameters for tracking without saving the data. After that, the camera capture is restarted while keeping the tracking configuration, followed by recording.

### A custom-built light-sheet microscopy system

Sardine orchestrates complex systems with highly coupled hardware parts and simplifies the prototyping process of such systems. To demonstrate this, we used Sardine to build a control application for a light-sheet microscope, as represented by the graph in Fig 7. The system contained several parts of hardware that had to be managed, including a data acquisition board for synchronizing hardware inputs and outputs, an electrically tunable lens, and two sCMOS (scientific Complementary Metal–Oxide–Semiconductor) cameras capable of producing data streams of 800 MB.s$^{-1}$ each. We designed abstractions to represent the light sheet and used those abstractions as system components. This system also incorporates a modified version of the previously described tail tracking system. The architecture of this software is schematized in its graph representation, with the intent of showing that even highly coupled hardware architectures can benefit from the modular approach that Sardine provides.

To control the microscope hardware, we built a model that uses waveforms as building blocks to generate synchronous output signals. This model is meant to give components a means to access data acquisition boards to control hardware parts with microsecond precision. It includes waveform synthesizers, a block-building mechanism for combining patterns of waveforms, and a sequencer that combines the blocks into a full output array provided to the hardware device. Additionally, this model includes support for virtual output channels, corresponding to software instructions executed synchronously at predetermined block positions in near real-time (Fig 8).

We provide code for operating some hardware components, including National Instruments data acquisition boards and Hamamatsu sCMOS cameras, and abstractions to represent optical systems.

## Alternatives

Several software solutions that share common spaces with Sardine have been developed, either as general-purpose or domain-specific solutions targeting the presented use cases, which follow a modular or graph-based design. While we think Sardine has a unique combination of features, alternative solutions exist that might be considered for different applications. Here, we discuss some of the most likely choices and how their features and use cases compare with our framework.

### General-purpose graph-based systems

Bonsai is an open-source framework for processing and controlling data streams. Its emphasis is on facilitating the implementation of experimental designs that depend on asynchronous data streams [19]. Bonsai includes a graphical user

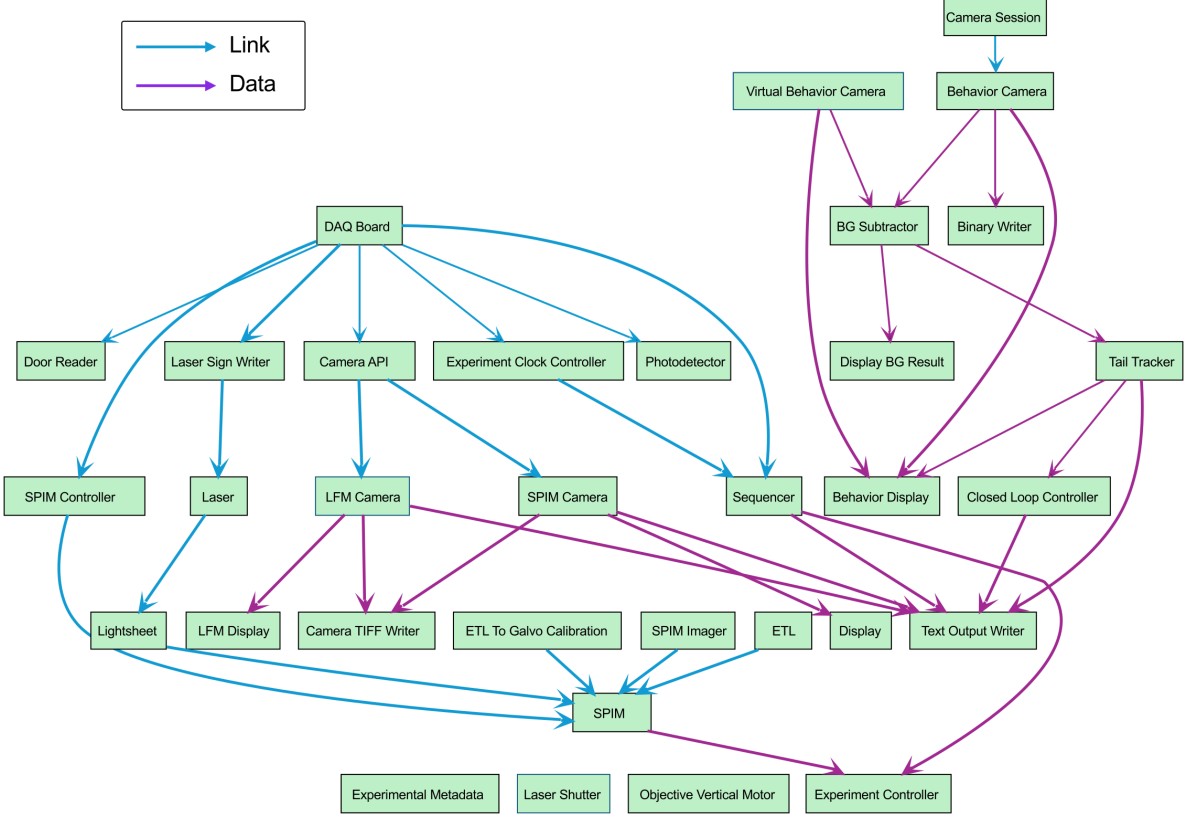

**Fig 7**. **Fleet graph diagram for an example microscope application.** Blue arrows represent the link graph edges, and purple arrows represent the data graph edges.

interface for building application flows from a collection of pre-built nodes. Through a drag-and-drop model, a user can build a workflow without needing any specific programming knowledge. The library of nodes is extensive, from basic data operations to imaging processing tools and support for standard hardware such as Arduino microcontrollers or USB cameras. It is extensible, as new nodes can be designed and integrated with the framework. Bonsai is also a reactive programming language where nodes use event callbacks to receive and process incoming data. Furthermore, Bonsai nodes can be built and maintained separately, simplifying code reuse. Since its introduction, Bonsai has grown a thriving community of users. Several user-made packages have been made available for the general community, including BonZeb [20], a set of Bonsai nodes for tracking and analyzing larval zebrafish behavior.

Bonsai, like Sardine, achieves asynchronicity through an event-driven approach. Bonsai leverages the Rx.NET framework [21], which is built to manipulate live data streams, and Sardine uses a lightweight implementation of a publisher-subscriber pattern.

Unlike Sardine, all Bonsai nodes communicate through event-based messages, and it is not possible to assert coupled dependencies between nodes that would allow for synchronous operations. Furthermore, Bonsai nodes aren't automatically protected against network failures: a stalling node causes the whole tree to be affected, potentially disrupting the system.

LabVIEW is a graphical general-purpose language for building data flows using a graphical user interface, which provides blocks (virtual instruments, or VIs) that fulfill logic functions or control pieces of hardware that can be wired together

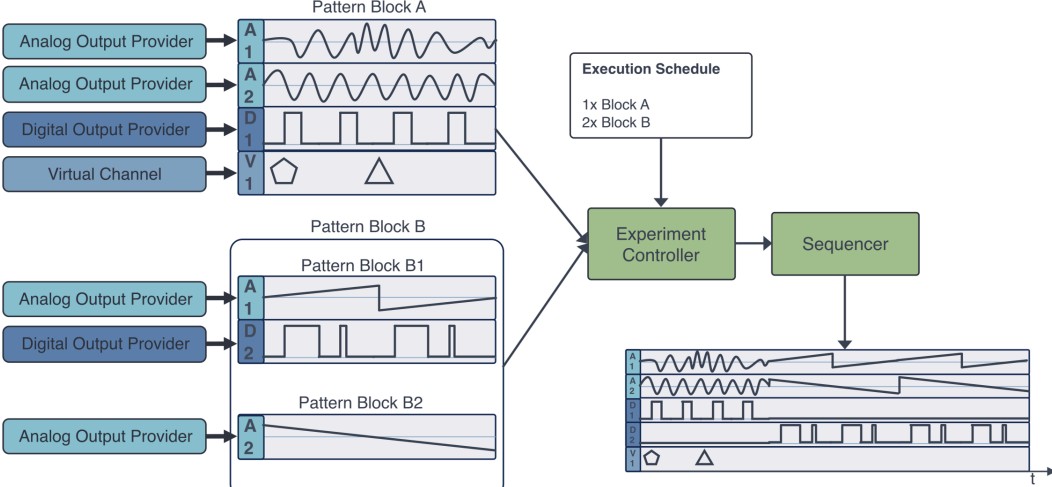

**Fig 8**. **A model for real-time control of hardware with Sardine**. A set of waveform synthesizers or other signal providers is associated with physical hardware channels (here represented as **A** for analog channels and **D** for digital channels) and virtual channels (represented as **V**) consisting of a series of instructions that the software will execute at given moments in time, forming pattern blocks. Then, an `Experiment Controller` component aggregates those blocks, together with an execution schedule provided by the user, and sends it to a `Sequencer` component, which builds the final waveforms sent to the corresponding hardware devices.

to control complex hardware systems [22]. In LabVIEW, a 'front-end' view for the user is created directly from VIs, allowing system control. LabVIEW employs complex algorithms for polling and maintaining the hardware status, which requires extensive computational overhead. LabVIEW is not an open-source solution: its licensing costs are high, and it is hard to predict how its internal algorithms will deal with multiple requests. Debugging LabVIEW workflows remains a complex task, as the amount of backend wiring quickly increases with the growing complexity of the system being modeled. This contrasts with the modular structure of Sardine, where each component can be modeled and debugged independently, contributing to long-term maintainability and scalability of the system. While recent solutions have been proposed to develop block-based graphical programming languages using open-sourced polling algorithms [23], whether these systems could handle large data streams is unclear.

Heron [24] is another open-source tool that allows developers to build custom nodes that can then be aggregated into a graph. However, this tool does not support asynchronous data communication nor include support for buffering data at the node level; instead, all modules are synchronized via a heartbeat system. This could pose significant challenges for running complex or time-critical systems, where the impact of delays could be considerable.

Falcon [25] and BRAND [26] are open-source software for executing arbitrary graphs of processing nodes at low latencies and asynchronously. The former runs a local multi-threaded system with nodes pushing their output into ring buffers, which subsequent nodes can read. In contrast, the latter runs several processes in parallel, which communicate by accessing data stored in a Redis database and can be applied to distributed computing models. Those solutions support tight closed-loop control and use memory streams to uncouple the input/outputs of each module. Being focused on closed-loop data processing, they are not designed to operate the acquisition hardware and do not natively integrate support for graphical user interfaces.

## Domain-specific solutions

While we have developed Sardine to support a wide range of demanding applications, there are alternative frameworks that have been purpose-built for some of the use cases highlighted here. These tools may offer convenient

application-specific features and configurations that make them attractive choices in those contexts. Nevertheless, we believe that Sardine provides unique advantages in terms of flexibility, extensibility, and general-purpose capacity that can be valuable when adapting to new experimental paradigms or combining diverse systems into a unified framework.

The most commonly used open-source software tool for controlling custom microscopes is μManager [27,28], which is a domain-specific application running on top of ImageJ [29] containing a collection of modules for controlling standard microscopy hardware (such as cameras, filter wheels, turrets, and illumination sources) and microscope control routines (calibration tools, phototargeting, hardware synchronization of components). It includes GUIs and provides access to all ImageJ image processing features.

Stytra [30] is an open-source software package that aims to cover all aspects of running behavioral experiments in larval zebrafish through a modular approach, including posture tracking and closed-loop control of visual stimulation. It also features hooks to the popular pose-tracking system DeepLabCut [31], and can be extended for use with other model systems.

improv [32] is a software platform that allows for the control of closed-loop experimental pipelines that implement online data analysis using established pipelines. For example, imaging data collected by a microscope is placed on a storage server, where improv routes the data through a set of actors performing the required computations, such as the extraction of regions of interest or correlation analysis. This tool allows users to characterize functional neural responses of larval zebrafish with up-to-the-minute feedback and use this information to perform adaptive optogenetics experiments. Similar to Falcon and BRAND, improv finds usage in performing closed-loop experiments.

## Performance benchmarks

We wanted to know how much computational overhead is required by Sardine. To obtain a quantitative measurement of the efficiency and robustness of Sardine, we prepared a test operator capable of fully loading the CPU by performing arbitrary mathematical operations. By measuring how long Sardine would take to process data using the test operator and feed downstream ($T_{framework}$), and knowing the reference time needed to run the test operator ($T_{ref}$), one can calculate the overhead of the framework $\frac{T_{framework} - T_{ref}}{T_{ref}}$. To estimate the processing time of Sardine, we created a Sardine application comprised of the following modules in sequence: (i) a source input generating requests at a given rate, (ii) a timestamper, (iii) the test operator, (iv) a second timestamper, (v) a time difference calculator, and (vi) a text writer to record the timestamps. Although our arguments for Sardine are based on available features rather than performance alone, we understand that it is important to show that performance is at least as good as other state-of-the-art methods. We compared Sardine with a similar event-driven open-source framework: Bonsai. We created a Bonsai application following the same layout: we built a custom Bonsai node running the same test operator to act as the load. We used the BenchmarkDotNet library to estimate the reference time needed to run the test operation.

To facilitate time measurements, we tuned the test operator to require several milliseconds to complete in the test machine. We settled on a set of 10 tests with different difficulty loads, with compute times ranging from 2 to 20 milliseconds. We set the frameworks to request test operations at a fixed rate of 50 Hz, and collected samples at each difficulty.

Both Sardine and Bonsai can run the test operator with no notable delay for small loads (Fig 9A). As expected, the reference times are always lower than those required by the frameworks, as no extra control operations or support features are available.

Our test shows that Sardine's performance is consistently higher than that of Bonsai's. Sardine could run at the fixed rate up until difficulty 9, and Bonsai could only run at the fixed rate up until difficulty 6.

Running Sardine at difficulty 10 led to one of Sardine's possible failure modes: incoming data accumulated while the source kept operating at its set rate of 50 Hz, and the processing time of each message kept increasing due to system overload: no CPU time was available to keep maintaining the framework, leading to an unstable state downstream of the test operator module.

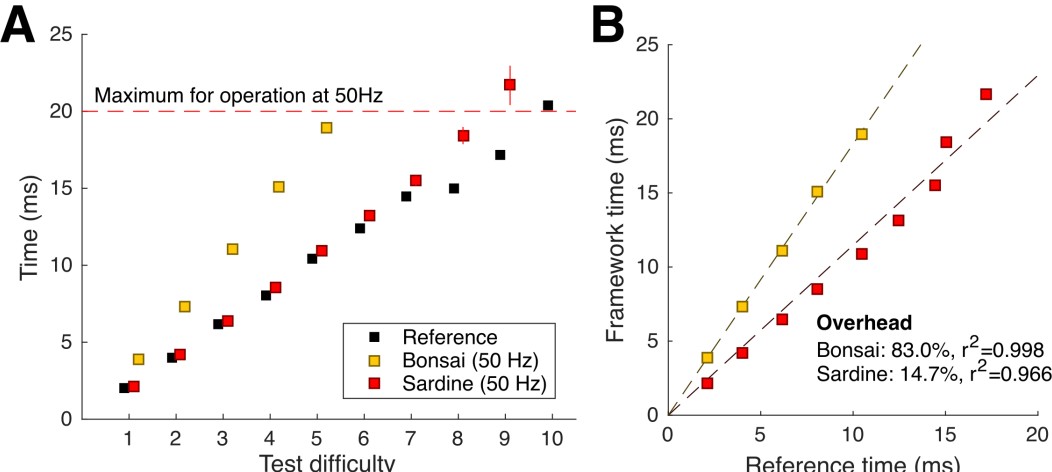

**Fig 9**. **Sardine performance, measured against Bonsai and a baseline reference**. **A** Time taken to complete one test operation for different difficulty levels. Dots indicate the mean, and bars indicate the standard deviation. The frameworks were set to request test operations at 50 Hz. **B** Overhead of Sardine and Bonsai, calculated as the percentage of extra time required to complete a test operation with a framework versus without one. Data points were fit using a simple linear regression with no intercept.

We estimated the percentage of overhead required to run the frameworks (Fig 9B).

We fitted the data using a simple linear regression, with no intercept ($y = ax$), and obtained an estimate for the framework's overhead. In this task, Sardine's estimated overhead (14.7%) is considerably lower than Bonsai's (83.0%).

## Discussion

We developed Sardine, an open-source framework built with `.NET` that creates and manages complex workflows in data acquisition systems. We have demonstrated the framework's usefulness by implementing software solutions to run high-throughput experimental setups in our lab and produce useful data, and showed how Sardine's two-layered component aggregator is used to provide fault-tolerance by asserting the working state of critical components.

Sardine's design paradigm asserts that a software solution built with it will have a well-defined level of coupling determined by its graph connections. Components can be developed independently, allowing for easy reuse. This approach simultaneously accelerates the processes of prototyping, testing, and maintaining software solutions. Adding to the framework, we have released access to a gallery of components that can be used with Sardine and serve as a base for developing further components. Notably, we have included components for reliably collecting experimental metadata and ensuring a repeatable experimental workflow, and components for controlling data acquisition boards through a block-building process combining waveform synthesizers and a pattern sequencer.

While Sardine addresses several challenges of developing desktop applications for research, there is still room to grow. The current version of Sardine presents several caveats that deserve consideration. Firstly, while Sardine is built using `.NET 8`, meaning its core functionality is compatible across multiple operating systems, most of the UI was designed in WPF, which is only available for Windows. Recent cross-platform alternatives, such as AvaloniaUI [33] or newer .NET frameworks like .NET MAUI [34], could replace WPF, allowing built-in Sardine GUI to be available on most operating systems. Despite this limitation, the Sardine framework itself remains fully cross-platform, ensuring its core functionality can be utilized across diverse environments. Secondly, we have found that incorporating circular buffers into components that would generate large messages dramatically reduces the load on the garbage collector. We are planning an update where data sources use circular buffers by default as their output.

                                                                        

Another challenge Sardine faces is assessing the framework's performance when used in applications with long lifecycles. Additionally, it's essential to understand how the core mechanisms of Sardine may affect long-term code reusability in practice. Although this is a complex metric to quantify, we have successfully maintained and adapted software using Sardine for several years and across numerous experimental setups, without needing to rewrite large portions of code.

Although the examples we've shown are applied in neuroscience, Sardine's core structure is domain agnostic, so it will prove equally valuable for orchestrating applications in other domains. We aim to develop and maintain Sardine further as a general framework for controlling custom experimental setups and data acquisition systems, and promote its wider adoption.

## Acknowledgments

We thank Simão Reis for the valuable discussions and for reviewing the manuscript text. We thank the various members of the Vision to Action laboratory who used and tested Sardine by using applications built with it and who provided critical feedback.

## Author contributions

**Conceptualization:** A. Lucas Martins, Alexandre Laborde, Michael B. Orger.

**Formal analysis:** A. Lucas Martins, Alexandre Laborde.

**Funding acquisition:** A. Lucas Martins, Michael B. Orger.

**Investigation:** A. Lucas Martins, Alexandre Laborde.

**Methodology:** A. Lucas Martins, Alexandre Laborde.

**Project administration:** Michael B. Orger.

**Software:** A. Lucas Martins, Alexandre Laborde.

**Supervision:** Michael B. Orger.

**Visualization:** A. Lucas Martins.

**Writing – original draft:** A. Lucas Martins.

**Writing – review & editing:** A. Lucas Martins, Alexandre Laborde, Michael B. Orger.

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
