## [Decision Letter · Decision Letter 0]

3 Oct 2025

PONE-D-25-42410Sardine: a modular framework for developing data acquisition and near real-time analysis applicationsPLOS ONE

Dear Dr. Orger,

Thank you for submitting your manuscript to PLOS ONE. After careful consideration, we feel that it has merit but does not fully meet PLOS ONE’s publication criteria as it currently stands. Therefore, we invite you to submit a revised version of the manuscript that addresses the points raised during the review process.

We found the work to be technically sound and a valuable contribution to both neuroscience and software engineering. Sardine addresses an important need for modular, scalable, and reusable frameworks, and the examples presented demonstrate clear utility.

However, before acceptance, some revisions are needed. In particular, please include quantitative performance benchmarks (e.g., latency, CPU/memory usage, throughput) and, where possible, compare them with existing frameworks such as Bonsai or LabVIEW. The novelty of the two-layer architecture should be emphasized more explicitly, and further clarification is needed on cross-platform applicability, vessel management under high-throughput conditions, and user accessibility. We also encourage you to improve figure resolution and clarity, define acronyms at first use, and streamline some sections to enhance readability.

These revisions are modest and primarily aimed at improving clarity, accessibility, and impact. We therefore invite you to submit a revised version, after which the manuscript will be suitable for publication.

We look forward to receiving your revised manuscript.

Kind regards,

Inbakandan Dhinakarasamy, Ph.D

Academic Editor

PLOS ONE

“This work was financed by funding to Michael Orger's lab from the European Research Council (Consolidator Grant, Neurofish-DLV-773012), the Champalimaud Foundation, the Volkswagen Stiftung Life? Initiative (A126151) and through national funds from the Portuguese FCT - Foundation for Science and Technology, I.P., under the projects PTDC/NEU-SCC/5221/2014 and 2023.14873.PEX. A. Lucas Martins received funding from the Portuguese Fundação para a Ciência e Tecnologia (FCT) through the PhD fellowships SFRH/BD/129843/2017 and COVID/BD/152726/2022.”

Additional Editor Comments:

This manuscript presents Sardine, an open-source modular framework developed in the .NET environment to support hardware control, data acquisition, and near real-time analysis in experimental sciences, with particular emphasis on neuroscience applications. The framework is motivated by well-recognized challenges in the field, namely the unsustainability of custom-built experimental systems, the difficulty of reusing them across laboratories, and the lack of scalable, fault-tolerant, and domain-agnostic solutions. The authors describe a two-layer modular architecture, provide examples of its application in zebrafish posture tracking and light-sheet microscopy, and compare Sardine with existing frameworks such as Bonsai, LabVIEW, µManager, and Stytra. The manuscript is technically sound, well structured, and offers both conceptual insights and practical demonstrations, making it a valuable contribution to the communities of software engineering and experimental neuroscience.

While the manuscript is of high quality, there remain some areas where clarification and refinement would strengthen its contribution. A recurring theme across the reviews is the need for quantitative performance benchmarks. Including latency distributions, throughput, CPU and memory usage, and error rates, ideally in a comparative table with other frameworks such as Bonsai or LabVIEW, would provide concrete evidence of Sardine’s efficiency and robustness. Similarly, while the dual-layer architecture is central to Sardine’s design, its novelty and advantages could be emphasized more clearly in both the introduction and the discussion. In particular, the differences in long-term maintainability and scalability compared with Bonsai’s asynchronous model and LabVIEW’s tightly coupled approach deserve fuller elaboration.

Another important point relates to cross-platform usability. At present, the GUI is Windows-specific through .NET/WPF. The manuscript would benefit from further discussion of how Sardine can be used by non-Windows users, for example through command-line operation, headless mode, integration with Python interfaces, or containerization solutions such as Docker. A brief roadmap outlining possible directions for cross-platform support would broaden the appeal of the framework considerably.

Some technical aspects would also benefit from additional detail. The description of vessel abstractions and state management is clear, but more information is needed on how Sardine handles potential race conditions when multiple vessels attempt to access a shared resource, and what the performance overhead is of vessel creation and dynamic reloading in high-throughput experiments. Similarly, although the examples provided are compelling, the results could be expanded to include quantitative performance outcomes for zebrafish posture tracking and microscopy applications, together with descriptions of how the system performs under stress conditions such as heavy CPU load or unexpected crashes.

Accessibility is another important factor. While the manuscript notes that Sardine is hosted on GitHub, providing readers with simple demo scripts or tutorials that can be run without specialized hardware would make it easier for the community to engage with and adopt the framework. At the same time, future plans for improving user experience and expanding multilingual support (for example in Spanish, Turkish, or Chinese) would help to establish Sardine as a globally relevant tool.

In terms of presentation, the manuscript would benefit from modest shortening to maintain reader engagement. Certain terms such as “freightering” and “invalidator” could be replaced with more standard software engineering terminology. Figures 2–4, although informative, appear dense and of low resolution; clearer rendering, more intuitive labeling, and color coding would help make vessel states and data graphs easier to follow. Finally, care should be taken to ensure that all acronyms (such as WPF and sCMOS) are defined at first mention.

In summary, the manuscript is a significant and timely contribution that successfully addresses challenges in developing modular, scalable, and reusable frameworks for experimental research. The work demonstrates both technical rigor and practical relevance. With the inclusion of benchmarking data, clarification of architectural novelty, improved guidance on cross-platform use, and modest improvements to figures and presentation, the manuscript will be ready for publication.

Reviewers' comments:

Reviewer's Responses to Questions

**Comments to the Author**

1. Is the manuscript technically sound, and do the data support the conclusions?

Reviewer #1: Yes

Reviewer #2: Yes

Reviewer #3: Yes

2. Has the statistical analysis been performed appropriately and rigorously?

Reviewer #1: Yes

Reviewer #2: Yes

Reviewer #3: N/A

3. Have the authors made all data underlying the findings in their manuscript fully available?

Reviewer #1: Yes

Reviewer #2: Yes

Reviewer #3: Yes

4. Is the manuscript presented in an intelligible fashion and written in standard English?

Reviewer #1: Yes

Reviewer #2: Yes

Reviewer #3: Yes

5. Review Comments to the Author

Reviewer #1: This study introduces Sardine, a software framework developed for use in experimental sciences, particularly in data-intensive and hardware-focused fields such as neuroscience. Sardine was developed within the .NET ecosystem and aims to facilitate the establishment of hardware control and near real-time data processing systems by operating in a modular structure. The software features fault-tolerant architecture, data queue management, module independence, dual-layer module integration, interface creation tools, and remote access. Examples of Sardine's use include neuroscience experiments such as zebrafish tracking and light-transmission microscope control. The work stems from practical problems such as the unsustainability of hardware-software integrations used in experimental research and the difficulty of reusing custom systems. Sardine is a framework developed as a solution to these problems. Its motivation is clear, powerful, and field-based. It combines hardware control with real-time data processing, creates fault-tolerant systems, and offers an open-source platform.

Although software-based, it offers significant methodological contributions to the design of experimental systems in disciplines such as neuroscience. Comparisons with existing solutions (Bonsai, LabVIEW, µManager, etc.) are clear. However, I would like to kindly offer the following suggestions to the authors:

1- Benchmarking is lacking. No direct performance comparison has been made with alternatives such as Bonsai or LabVIEW. A Benchmarking Table is recommended, and Sardine's comparative performance data with other systems should be used.

2- Evaluation of user experience/learning curve is limited and should be provided in detail. Additionally, further content could be provided on advanced customizations in GUI design.

3- Real-time test results could be more detailed (e.g., CPU load, crash scenarios, resource usage). Also, a plan or roadmap should be shared for Docker/Cross-platform support (currently limited to .NET/WPF).

4- Support for multiple languages could be added; common languages should be added (e.g., Spanish, Turkish, and Chinese). I recommend adding them for global distribution.

My polite suggestion is to accept with minor revisions.

Reviewer #2: In the manuscript entitled "Sardine: a modular framework for developing data acquisition and near real-time analysis applications" authors presented the detailed structure and execution of Sardine open-source modular software built in .NET environment, for controlling experimental setups and real-time data acquisition/analysis. The manuscript is technically sound and provides both conceptual and practical insights into the framework. It is a well structured presentation with detailed examples and comparative analysis with alternative solutions. Overall, this is a valuable contribution to the field of software engineering, with potential broader applications. However, still there is a scope for further improvement by addressing the following suggestions and comments to improve clarity and readability.

1. Through the developed software Sardine provides alternate solution to the existing frameworks (e.g., Bonsai, LabVIEW, µManager, Stytra), the novelty of Sardine’s "two-layer architecture" (link layer + data layer) may be more explicitly emphasized in the Introduction and Discussion.

2. Authors may clarify how Sardine differs with respect to Bonsai’s asynchronous model and LabVIEW’s coupling in terms of long-term maintainability and scalability (other than error handling).

3. Currently, Sardine GUIs limits usage on Windows only. Authors may describe how non-Windows users can still utilize Sardine (e.g., CLI-only operation, headless mode, or integration with Python frontends).

4. The vessel abstraction and state management (IsOnline, IsActive, ExecuteCall) are clearly explained, but some technical details require more elaboration. For example:

• How does Sardine handle race conditions when multiple vessels try to access a shared resource?

• What is the overhead of vessel creation and dynamic reloading in high-throughput experiments?

5. Though the availability of Sardine on GitHub repositories is mentioned, authors may provide some working examples (e.g., a reproducible demo script or tutorial) in the manuscript that allows readers to test Sardine immediately without specialized hardware.

6. While the zebrafish posture tracking and light-sheet microscopy examples are appealing, quantitative performance metrics may be provided for posture tracking, report actual latency distributions, error rates, and system throughput.

Minor Comments:

1. The current manuscript is a valuable contribution but the length of the manuscript diverts the readers interest. Authors are advised to shorten the manuscript with more focus to attach the readers interest.

2. Usage of some words like "freightering", "invalidator" may be replaced with more standard terminology.

3. Figures 2–4 are informative but somewhat dense and with low resolution. Use clearer color coding or labels to make vessel states and data/link graphs more intuitive.

4. Ensure all acronyms (e.g., WPF, sCMOS) are defined at first use.

Reviewer #3: The manuscript provides valuable contributions to both the software engineering and experimental neuroscience communities by offering a domain-agnostic, modular, and scalable solution for real-time data acquisition and analysis. To further strengthen the article, the authors could include quantitative performance benchmarks (e.g., latency, memory usage, CPU load) in the Methods or Use Cases sections, provide diagrams or figures for all architectural components, and emphasize cross-platform GUI alternatives for broader applicability.

6. PLOS authors have the option to publish the peer review history of their article (what does this mean?). If published, this will include your full peer review and any attached files.

Reviewer #1: No

Reviewer #2: No

Reviewer #3: No

---

## [Author Response · Author response to Decision Letter 1]

3 Dec 2025

Thank you for the positive and constructive feedback provided on our paper. Below we give point by point responses to the editorial and reviewers requests.

We have verified that the manuscript and file names adhere to PLOS ONE’s style guidelines.

“This work was financed by funding to Michael Orger's lab from the European Research Council (Consolidator Grant, Neurofish-DLV-773012), the Champalimaud Foundation, the Volkswagen Stiftung Life? Initiative (A126151) and through national funds from the Portuguese FCT - Foundation for Science and Technology, I.P., under the projects PTDC/NEU-SCC/5221/2014 and 2023.14873.PEX. A. Lucas Martins received funding from the Portuguese Fundação para a Ciência e Tecnologia (FCT) through the PhD fellowships SFRH/BD/129843/2017 and COVID/BD/152726/2022.”

We have provided the statement above in the cover letter.

We have not, to our knowledge, cited a retracted study, or any new citations requested by the reviewers.

This manuscript presents Sardine, an open-source modular framework developed in the .NET environment to support hardware control, data acquisition, and near real-time analysis in experimental sciences, with particular emphasis on neuroscience applications. The framework is motivated by well-recognized challenges in the field, namely the unsustainability of custom-built experimental systems, the difficulty of reusing them across laboratories, and the lack of scalable, fault-tolerant, and domain-agnostic solutions. The authors describe a two-layer modular architecture, provide examples of its application in zebrafish posture tracking and light-sheet microscopy, and compare Sardine with existing frameworks such as Bonsai, LabVIEW, µManager, and Stytra. The manuscript is technically sound, well structured, and offers both conceptual insights and practical demonstrations, making it a valuable contribution to the communities of software engineering and experimental neuroscience.

We sincerely thank the editor for handling our manuscript and for the constructive and detailed feedback from the reviewers. We are very happy that our work was considered to make a valuable contribution to the fields of software engineering and experimental neuroscience and to be technically sound.

While the manuscript is of high quality, there remain some areas where clarification and refinement would strengthen its contribution. A recurring theme across the reviews is the need for quantitative performance benchmarks. Including latency distributions, throughput, CPU and memory usage, and error rates, ideally in a comparative table with other frameworks such as Bonsai or LabVIEW, would provide concrete evidence of Sardine’s efficiency and robustness.

Thank you for this feedback. Although our arguments for using Sardine are based predominantly on available features rather than performance alone, we understand that it is important to show that performance is at least as good or better than other state of the art methods. We have run performance benchmarks on the framework, providing quantitative measurements of the latency distribution and the overhead of Sardine during a test operation under different CPU loads, increasing the stress to measure a failure point. We compared Sardine with one of the most comparable open-source frameworks, Bonsai, and showed that Sardine compares very favorably, showing considerably lower overhead in this test. The results were compiled and added to the manuscript in a new section called “Performance benchmarking”.

Similarly, while the dual-layer architecture is central to Sardine’s design, its novelty and advantages could be emphasized more clearly in both the introduction and the discussion.

We added the following paragraph to the introduction, reinforcing the main advantage of running a dual-layer architecture: “Components are connected in a novel two-layer network represented by two graphs: one defining hierarchical dependencies and the other defining data flows. By separating connections into critical dependencies and non-critical messaging, a system built with Sardine can operate even in the presence of perturbations, as malfunctioning components and their hierarchical dependencies can be safely terminated by the framework.”

We added the following sentence to the first paragraph of the discussion, emphasizing the main advantage of the two-layer architecture: “, and showed how Sardine's two-layered component aggregator is used to provide fault-tolerance by asserting the working state of critical components.”

In particular, the differences in long-term maintainability and scalability compared with Bonsai’s asynchronous model and LabVIEW’s tightly coupled approach deserve fuller elaboration.

Sardine and Bonsai follow similar approaches to asynchronicity, which should not affect maintainability. We clarified the difference in the asynchronous patterns used by Bonsai and Sardine: “Bonsai, like Sardine, achieves asynchronicity through an event-driven approach. Bonsai leverages the Rx.NET framework, which is built to manipulate live data streams, and Sardine uses a lightweight implementation of a publisher-subscriber pattern.”, and added the note “Furthermore, Bonsai nodes can be built and maintained separately, simplifying code reuse.”. We also added a reference to the Rx.NET framework website (Microsoft. ReactiveX, 2011. Available in https://reactivex.io/).

We added a note reinforcing how Sardine’s modular structure increases its long-term maintainability and scalability: “ This contrasts with the modular structure of Sardine, where each component can be modeled and debugged independently, contributing to long-term maintainability and scalability of the system”.

Another important point relates to cross-platform usability. At present, the GUI is Windows-specific through .NET/WPF. The manuscript would benefit from further discussion of how Sardine can be used by non-Windows users, for example through command-line operation, headless mode, integration with Python interfaces, or containerization solutions such as Docker. A brief roadmap outlining possible directions for cross-platform support would broaden the appeal of the framework considerably.

Since the core of Sardine is written in .NET 8, which already has native cross-platform support, any non-Windows user can already use the Sardine core architecture. In the discussion, we suggest steps that any user with sufficient interest can take to produce compatible GUIs to use Sardine in non-Windows environments: “Recent cross-platform alternatives, such as AvaloniaUI or newer .NET frameworks like .NET MAUI could replace WPF, allowing built-in Sardine GUI to be available on most operating systems”.

Some technical aspects would also benefit from additional detail. The description of vessel abstractions and state management is clear, but more information is needed on how Sardine handles potential race conditions when multiple vessels attempt to access a shared resource, and what the performance overhead is of vessel creation and dynamic reloading in high-throughput experiments.

We have added a description on how multiple accesses to a shared resource are resolved to the manuscript: “Furthermore, since messages are placed in the inbox queue before processing, if multiple vessels try to communicate with the same downstream node, their messages will be added to the queue and processed sequentially, avoiding collisions in accessing reserved resources. Furthermore, since all messages carry a unique ID, messages can be uniquely resolved and ordered by the user as appropriate if such behavior is required”.

Since vessel creation and dynamic reloading are not expected to occur at high rates during the experiment, the practical performance cost of these actions should be negligible, as any temporary delays will be dampened by the modules' inbox queues. We have added a note describing this design consideration to the manuscript: “The inbox queue also dampens any temporary delays caused by vessel creation or their dynamic reloading.”

Similarly, although the examples provided are compelling, the results could be expanded to include quantitative performance outcomes for zebrafish posture tracking and microscopy applications, together with descriptions of how the system performs under stress conditions such as heavy CPU load or unexpected crashes.

In-depth performance tests for concrete applications, which depend on specific hardware parts such as cameras or data acquisition systems, may not represent the framework accurately; therefore, we have designed quantitative benchmarks, described above, that can demonstrate performance in a more neutral setting. Nevertheless, we added information on the average time required for our machine to compute fish posture per camera frame to the “Use Cases” section of the manuscript: “This simple tracker is efficient, computing tail posture in under 150 microseconds per frame, and making it robust to other stresses on system performance”.

We have added a description of the framework performance under stress conditions and failure modes in the new manuscript section “Performance benchmarks”.

Accessibility is another important factor. While the manuscript notes that Sardine is hosted on GitHub, providing readers with simple demo scripts or tutorials that can be run without specialized hardware would make it easier for the community to engage with and adopt the framework.

To enhance accessibility, we have made a demo software for an online zebrafish tail tracker, which does not require specialized hardware but instead uses a fish tail recording that is replayed by a virtual camera component, and is publicly available on GitHub (https://github.com/orger-lab/sardine-components). It is accompanied by a step-by-step tutorial on how to install and run the demo, and a link to this repository is available in the Sardine readme.

At the same time, future plans for improving user experience and expanding multilingual support (for example in Spanish, Turkish, or Chinese) would help to establish Sardine as a globally relevant tool.

As we do not have the knowledge to properly translate or validate translations into most languages, we believe that our providing multilingual translations of the Sardine documentation would not effectively expand its reach. Moreover, due to the ease of modern translation tools, such as those enabled by large language models, it should be possible for any user to generate their own translation by copy-pasting the documentation into such a tool (e.g. ChatGPT) to translate into their own language. Furthermore, as an open-source framework, any interested user could modify Sardine and add their own translation to the framework.

In terms of presentation, the manuscript would benefit from modest shortening to maintain reader engagement.

We appreciate the suggestion and have made efforts to edit the language of the manuscript to make it more concise. To keep the manuscript size manageable, even in light of the proposed clarifications and additions, we have substantially reduced the length of the introduction section, removing some redundancy, while maintaining the more important explanations of the methods and benchmarking.

Certain terms such as “freightering” and “invalidator” could be replaced with more standard software engineering terminology.

We replaced references to “freighting” in the manuscript by an explanation that the Freighter class is responsible for instantiating Vessels. We kept “invalidation” as we believe it correctly portrays the meaning of rendering an object as invalid/needing an update, and it is a term that sees use in other software engineering applications (e.g. UI element invalidation, cache invalidation). To clarify this meaning, we adapted the manuscript with the following sentence: “The invalidator properly marks the component as invalid in cases of termination or failure”.

Figures 2–4, although informative, appear dense and of low resolution; clearer rendering, more intuitive labeling, and color coding would help make vessel states and data graphs easier to follow.

We have adapted the colors of the vessel diagrams and data graphs to make the text easier to read and follow. We have also re-rendered all figures with improved resolution.

Finally, care should be taken to ensure that all acronyms (such as WPF and sCMOS) are defined at first mention.

The acronyms WPF, UI, GUI, and sCMOS are now defined at first mention. The acronym ROI was replaced by its definition.

In summary, the manuscript is a significant and timely contribution that successfully addresses challenges in developing modular, scalable, and reusable frameworks for experimental research. The work demonstrates both technical rigor and practical relevance. With the inclusion of benchmarking data, clarification of architectural novelty, improved guidance on cross-platform use, and modest improvements to figures and presentation, the manuscript will be ready for publication.

We have carefully addressed all of the editor and reviewers' comments and believe the manuscript has been significantly strengthened as a result of their suggestions, better demonstrating the value of Sardine as a modular framework for developing data acquisition and near real-time analysis applications. In particular, we have added performance benchmark data, explained in the manuscript how Sardine’s architecture differs from other state of the art solutions, clarified how cross-platform use is currently being supported by Sardine, and re-rendered all figures at a higher resolution. We believe that the revised manuscript now meets the standards for publication in PLOS ONE.

5. Review Comments to the Author

Reviewer #1: This study introduces Sardine, a software framework developed for use in experimental sciences, particularly in data-intensive and hardware-focused fields such as neuroscience. Sardine was developed within the .NET ecosystem and aims to facilitate the establishment of hardware control and near real-time data processing systems by operating in a modular structure. The software features fault-tolerant architecture, data queue management, module independence, dual-layer module integration, interface creation tools, and remote access. Examples of Sardine's use include neuroscience experiments such as zebrafish tracking and light-transmission microscope control. The work stems from practical problems such as the unsustainability of hardware-software integrations use

---

## [Decision Letter · Decision Letter 1]

5 Feb 2026

Sardine: a modular framework for developing data acquisition and near real-time analysis applications

PONE-D-25-42410R1

Dear Dr. Michael Orger,

We’re pleased to inform you that your manuscript has been judged scientifically suitable for publication and will be formally accepted for publication once it meets all outstanding technical requirements.

Kind regards,

Inbakandan Dhinakarasamy, Ph.D

Academic Editor

PLOS One

Additional Editor Comments (optional):

The revised manuscript presents a technically sound, well-motivated, and practically validated software framework that addresses an important challenge in experimental research software development. The authors have satisfactorily addressed all reviewer comments, and the clarity of presentation has improved, particularly in describing the modular architecture, fault tolerance, and real-world applicability. The separation of control dependencies and data flow is well justified and effectively demonstrated through relevant use cases. Overall, the work reflects strong software engineering principles, offers clear utility to the research community, and meets the standards of rigor and clarity expected by the journal. In its present form, the manuscript is suitable for acceptance for publication.

Reviewers' comments:

Reviewer's Responses to Questions

**Comments to the Author**

1. If the authors have adequately addressed your comments raised in a previous round of review and you feel that this manuscript is now acceptable for publication, you may indicate that here to bypass the “Comments to the Author” section, enter your conflict of interest statement in the “Confidential to Editor” section, and submit your "Accept" recommendation.

Reviewer #2: All comments have been addressed

Reviewer #3: All comments have been addressed

Reviewer #4: All comments have been addressed

2. Is the manuscript technically sound, and do the data support the conclusions?

Reviewer #2: Yes

Reviewer #3: Yes

Reviewer #4: Yes

3. Has the statistical analysis been performed appropriately and rigorously?

Reviewer #2: Yes

Reviewer #3: Yes

Reviewer #4: Yes

4. Have the authors made all data underlying the findings in their manuscript fully available?

Reviewer #2: Yes

Reviewer #3: Yes

Reviewer #4: Yes

5. Is the manuscript presented in an intelligible fashion and written in standard English?

Reviewer #2: Yes

Reviewer #3: Yes

Reviewer #4: Yes

6. Review Comments to the Author

Reviewer #2: I appreciate the authors for revising the manuscript by considering the comments and suggestions given by the reviewers. The manuscript is sound enough to be accepted for publication.

Reviewer #3: This manuscript presents Sardine, a modular and fault-tolerant framework for developing near real-time data acquisition and control applications. The work addresses a well-recognized challenge in experimental research: the difficulty of building flexible, maintainable, and robust software systems that integrate heterogeneous hardware while supporting concurrent data processing.

The proposed architecture, based on the separation of hierarchical dependencies (link graph) from asynchronous data flow (data graph), is well motivated and clearly articulated. The vessel abstraction provides an effective mechanism for encapsulating component lifecycle, state management, and failure handling, and the framework demonstrates careful attention to concurrency, robustness, and software sustainability. The design reflects sound software engineering principles and a strong understanding of the practical constraints of desktop operating systems.

The manuscript is clearly written, logically structured, and accessible to both software engineers and experimental researchers. The inclusion of detailed code examples, configuration descriptions, and GUI integration workflows is particularly valuable and enhances the manuscript’s utility as both a research article and a reference resource. The presented use cases, especially those involving high-throughput behavioral tracking and microscopy control, convincingly demonstrate the framework’s applicability and scalability in demanding real-world scenarios.

The performance claims are supported by descriptive metrics appropriate for a software systems paper, and the absence of formal statistical hypothesis testing is reasonable given the nature of the contribution. The authors have made the framework and associated code openly available, supporting transparency and reproducibility. No concerns regarding research ethics, data availability, or dual publication are apparent.

I have only minor suggestions for improvement. The authors may wish to (i) more explicitly clarify the scope and current limitations of the framework, particularly regarding platform-specific UI components; (ii) provide brief contextual details for reported performance figures (e.g., typical hardware or measurement conditions); and (iii) slightly strengthen the comparative discussion with existing frameworks to help readers situate Sardine within the broader ecosystem of modular and graph-based systems.

Overall, this is a strong and timely contribution that will be of interest to researchers developing complex experimental systems. I recommend acceptance with minor revisions.

Reviewer #4: The work is technically excellent and publication-worthy. The Author addressed the reviewers’ concern too. Few sentences are Lengthy and could be split for readability. Overall, the work is technically strong, well motivated, and practically validated, making it a solid contribution to research

7. PLOS authors have the option to publish the peer review history of their article (what does this mean?). If published, this will include your full peer review and any attached files.

Reviewer #2: No

Reviewer #3: No

Reviewer #4: **Yes:** JABEZ J

---

## [Editor Report · Acceptance letter]

PONE-D-25-42410R1

PLOS One

Dear Dr. Orger,

I'm pleased to inform you that your manuscript has been deemed suitable for publication in PLOS One. Congratulations! Your manuscript is now being handed over to our production team.

Kind regards,

on behalf of

Dr. Inbakandan Dhinakarasamy

Academic Editor

PLOS One